# Mobilization practices in the ICU: A nationwide 1-day point- prevalence study in Brazil

**Karina T. Timenetsky**[1]\*, **Ary Serpa Neto**[1,2,3], **Murillo S. C. Assunção**[1],
**Leandro Taniguchi**[3,4], **Raquel A. C. Eid**[1], **Thiago D. Corrêa**[1,5], on behalf of the e-MOTION group[¶]

**1** Department of Critical Care Medicine, Hospital Israelita Albert Einstein, São Paulo, São Paulo, Brazil,
**2** Department of Intensive Care, Academic Medical Center, Amsterdam, The Netherlands, **3** Department of
Pneumology, Hospital das Clínicas HCFMUSP, Faculdade de Medicina, Universidade de São Paulo, São
Paulo, São Paulo, Brazil, **4** Department of Intensive Care, Hospital Sírio Libanês, São Paulo, São Paulo,
Brazil, **5** Brazilian Research in Intensive Care Network (BRICNet), São Paulo, Brazil

¶ Membership of the complete list of investigators in the e-MOTION study is provided in the
Acknowledgments.
\* karina.timenetsky@einstein.br

pone.0230971

Milan, ITALY

**Data Availability Statement:** All relevant data are
within the manuscript and its Supporting
Information files.

## Abstract

### Background

Mobilization of critically ill patients is safe and may improve functional outcomes. However,
the prevalence of mobilization activities of ICU patients in Brazil is unknown.

### Methods

A one-day point prevalence prospective study with a 24-hour follow-up period was con-
ducted in Brazil. Demographic data, ICU characteristics, prevalence of mobilization activi-
ties, level of patients' mobilization, and main reasons for not mobilizing patients were
collected for all adult patients with more than 24hs of ICU stay in the 26 participating ICUs.
Mobilization activity was defined as any exercise performed during ICU stay.

### Results

In total, 358 patients were included in this study. Mobilization activities were performed in
87.4% of patients. Patients received mobilization activities while under invasive mechanical
ventilation (44.1%), noninvasive ventilation (11.7%), or without any ventilatory support
(44.2%). Passive exercises were more frequently performed [46.5% in all patients; 82.3% in
mechanically ventilated patients]. Mobilization activities included in-bed exercise regimen
(72.2%). Out-of-bed mobility was reported in 39.9% of mobilized patients, and in 16.3% of
patients under invasive mechanical ventilation. The presence of an institutional early mobil-
ity protocol was associated with early mobilization (OR, 3.19; 95% CI, 1.23 to 8.22; p =
0.016), and with out-of-bed exercise (OR, 5.80; 95% CI, 1.33 to 25.30; p = 0.02).

**Funding:** The author(s) received no specific funding for this work.

**Competing interests:** The authors have declared that no competing interests exist.

## Conclusion

Mobilization activities in critically ill patients in Brazil was highly prevalent, although there was almost no active mobilization in the mechanically ventilated patients. Moreover, the presence of an institutional early mobility protocol was associated with a threefold higher chance of ICU mobilization during that day.

## Introduction

Muscle weakness with impaired physical function is a common complication of critical illness [1–4]. Muscle weakness can be defined as a "clinically detected weakness in patients in which there is no plausible etiology other than the critical illness itself" [5]. The reported incidence of muscle weakness in critically ill patients is between 30 to 50%, reaching up to 64% in septic patients [6]. The presence of muscle weakness has been associated with difficulties of weaning from mechanical ventilation, increased length of intensive care unit (ICU) and hospital stay, increased hospital costs, and long-term morbidity and mortality [7,8]. Early mobilization of critically ill patients may decrease the incidence of muscle weakness, and therefore improve outcomes [5–9].

Mobilization of critically ill patients is safe and feasible [10,11]. It improves, if applied early, independent physical status at hospital discharge [12,13], decreases the duration of mechanical ventilation [12], the number of days in delirium [12,13] and hospital length of stay [13], enhances recovery of functional exercise capacity [14], self-perceived functional status and muscle force at hospital discharge [14]. Nevertheless, recent studies have reported a low prevalence of mobilization activities in critically ill patients [15–18]. Moreover, mobilization is often limited to in-bed exercise [15–18]. For instance, Jolley and colleagues reported a 65% prevalence of mobilization activities in 42 ICUs in the United States of America [17]. In their study, non-mechanically ventilated patients were more likely to receive mobilization than mechanically ventilated patients, and approximately one third of mobilized patients received only passive activities [17]. A study performed in 11 Southern Brazilian ICUs reported a prevalence of 85% of mobilization activities in mechanically ventilated patients [18]. Nevertheless, this study evaluated only mechanically ventilated patients, and it reflects a regional pattern rather than a nationwide practice [18]. Therefore, we aimed to evaluate the prevalence of mobilization activities of critically ill patients in Brazilian ICUs through a nationwide one-day point prevalence study.

## Methods

### Design and setting

This was a 1-day prospective multicenter point prevalence study with a 24-hour follow-up period of mobilization activities of critically ill patients in Brazilian ICUs. The study was performed on June 29th, 2017. It was approved by the Hospital Israelita Albert Einstein's Ethics Committee (CAAE: 43545015.3.1001.0071), and each site obtained ethics approval for the study. Informed consent was obtained for all patients as requested by the ethics committee. The process used to obtain consent involved approaching the family member or patient when they met the study inclusion criteria, then explain the objective of the study and the possible uses of the information obtained from it. The person responsible for the study in each center was responsible for obtaining consent.

## Participants selection

Methods for recruitment of participating institutions included emailing members of the Brazilian Association of Intensive Care (Associação Brasileira de Medicina Intensiva, AMIB), announcements at national meetings and symposium, and emailing contacts and collaborators of each writing committee member. Adult patients ($\geq$ 18 years old) were eligible for inclusion if they were expected to stay at ICU for at least 24 hours. Exclusion criteria were patients with terminal disease or pregnancy.

Convenience sampling was used to include patients in the study.

## Data collection and study variables

Study data were collected and managed using Research Electronic Data Capture (REDCap) hosted at Hospital Israelita Albert Einstein [19]. The main investigators of each participating ICU completed an online survey about the hospital and the ICU characteristics, including type of hospital (public, private, and university), type of ICU (medical, surgical, mixed), number of ICUs beds, number of physiotherapists during a 6 hour shift, physiotherapist to patient ratio and nurse to patient ratio during a 6 hour shift, professional responsible for initiating patients' mobilization (physician, nurse, physiotherapist) and presence of institutional early mobility, sedation and delirium protocols. The full survey is presented in S1 Text.

Collected variables included demographics, comorbidities, ICU admission diagnosis, Sequential Organ Failure Assessment (SOFA) score [20], supportive therapy (need for vasopressors, invasive mechanical ventilation and noninvasive mechanical ventilation) during index ICU stay, type of ventilatory support, use of sedation (if receiving any type of sedation) and mobilization activities. The following patient variables were related to the study day: SOFA, supportive therapy, type of ventilatory support, sedation practices and mobilization activities.

## Mobilization activities

Mobilization activity was defined as any mobilization performed. Data on patients' mobility were collected during a 24-hour period in a single day (June 29[th], 2017). Prevalence of mobilization activities, the highest level of mobilization performed during the study day (in-bed or out-of-bed exercises), type of exercise performed (passive, assisted, active-assisted, active, and resisted exercises) and reasons for non performance of mobilization were collected. Contra indications for mobilization were considered as respiratory, cardiovascular, neurological or other considerations as described in the study published by Hodgson and colleagues [21]. Contra indication was considered present after the health care and research team reached consensus on this topic.

## Statistical analysis

Categorical variables are presented as absolute and relative frequencies. Continuous variables are presented as median with interquartile ranges (IQR).

Logistic regression models were used to evaluate factors associated with mobilization activity and with out-of-bed exercises. Predictors (independent variables) included into the logistic regression models were SOFA score [20], the use of invasive and noninvasive mechanical ventilation, the use of vasoactive drugs, type of hospital, type of ICU, number of physiotherapists per 6-hour shift, number of patients per physiotherapist, and the presence of institutional early mobility protocol. Multi-collinearity was checked for all variables. Results were presented as

odds ratio (OR) along with 95% confidence interval (95%CI). Statistical tests were two-sided. A $p < 0.05$ was considered statistically significant. All analyses were done in R (version 3.6.0).

## Results

### Characteristics of participating centers

A total of 26 ICUs participated in this study ([Fig 1]). The participating ICUs were located in Brazilian state capitals, mainly in the southeast (50% [13/26]), followed by the northeast (23% [6/26]), south (15.3% [4/26]), midwest (7.7% [2/26]), and north (3.8% [1/26]) of the country. Approximately half of them were ICUs located in public hospitals [53.8% (14/26)], followed by ICUs located in private [34.6% (9/26)] and university [11.5% (3/26)] hospitals. The majority of the participating ICUs (92.3%) were medical-surgical with a median (IQR) ICUs beds of 18 (10.0–31.5).

Most hospitals reported physiotherapy-initiated mobility [84.6% (22/26)]. A mobility protocol was reported in 57.7% (15/26) of ICUs while sedation and delirium protocols were reported in 50% (13/26) and 38.5% (10/26), respectively, of the ICUs.

During a six-hour shift, the median (IQR) number of physiotherapists and patient to physiotherapist ratio were 2.0 (1.0–3.2) and 8.0 (6.7–10.0), respectively. The median (IQR) patient to nurse ratio was 6.0 (4.0–8.5). The median (IQR) patient to nurse assistant ratio was 2 (2.0–3.0).

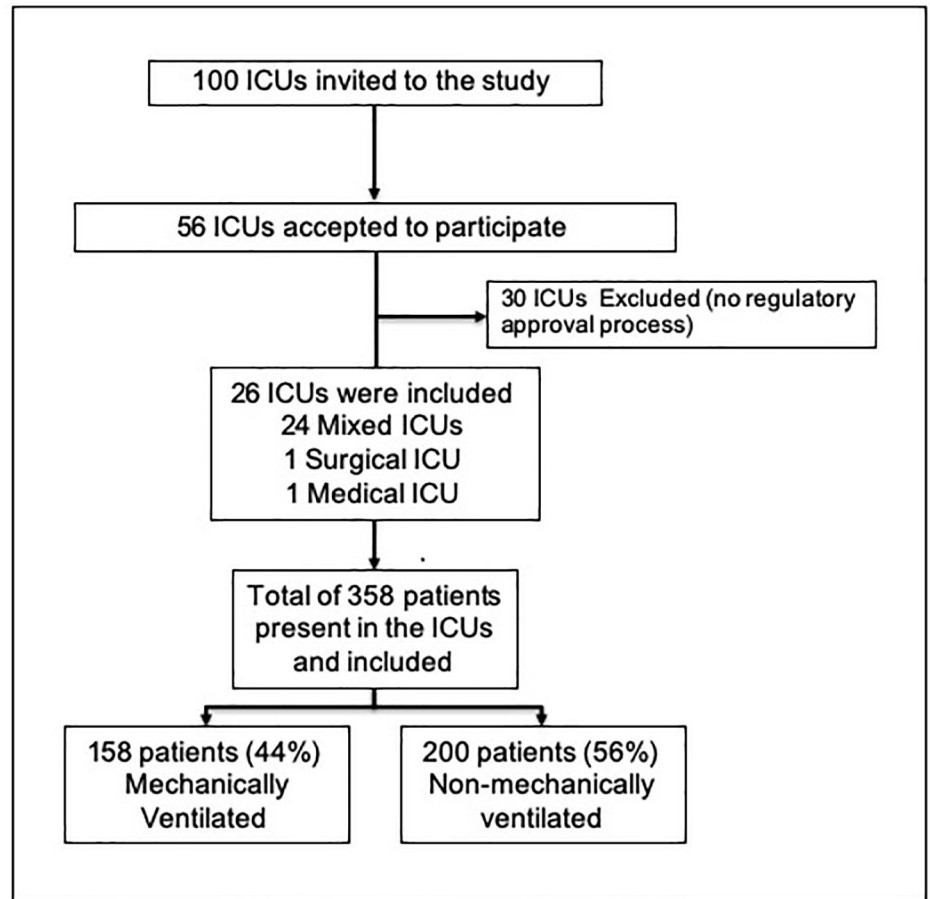

**Fig 1. Study flowchart.**

**Table 1. Baseline characteristics of study participants.**

| Characteristics | All Patients (n = 358) |
| --- | :---: |
| Age, years | 65 (53–76) |
| Men, n (%) | 190 (53) |
| SOFA score | 4.0 (2–7) |
| Reason for index ICU admission, n (%) | |
| Mixed | 332 (92.7) |
| Surgical | 19 (5.3) |
| Medical | 7 (2) |
| Reason for ICU admission, n (%) | |
| Respiratory | 120 (33.5) |
| Neurological | 68 (18.9) |
| Cardiologic | 61 (17.0) |
| Elective surgery | 50 (13.9) |
| Metabolic | 27 (7.5) |
| Gastric Intestinal | 24 (6.7) |
| Trauma | 6 (1.6) |
| Transplant | 2 (0.5) |
| Hospital category, n (%) | |
| Public | 184 (51.4) |
| Private | 130 (36.3) |
| University | 44 (12.3) |
| Sedation, n (%) | 80 (22.3) |
| Vasoactive drugs, n (%) | 100 (27.9) |
| Ventilatory support, n (%) | |
| Mechanical ventilation | 158 (44.1) |
| Noninvasive ventilation | 42 (11.7) |

Values represent median (IQR) or n (%). SOFA: Sequential Organ Failure Assessment.

## Studied population

The final sample included 358 patients (Fig 1). No patients were excluded from the analysis. The median (IQR) age was 65 (53–76) years, 53% of patients were male (Table 1). The prevalence of invasive and noninvasive mechanical ventilation during the study day was 44% and 11.7%, respectively (Table 1). Approximately one third of patients (27.9%) were receiving vasoactive drugs. The median (IQR) ICU length of stay to the study day was 6 (3–13) days for all included patients. A total of 117 (32.6%) patients were included within 72 hours of ICU admission.

## Mobilization activities

The overall prevalence of mobilization activities reported was 87.4% (313/358 patients). The prevalence of mobilization among mechanically and non-mechanically ventilated patients was 85.4% (135/158 patients) and 89% (178/200 patients), respectively (Table 2). The decision to initiate mobilization was most commonly related to the ICU physician evaluation [34.5% (108/313) of patients], followed by the physician and physiotherapist combined evaluation [29.0% (91/313) of patients], by the physiotherapist alone [22% (69/313) of patients], and a shared decision between physician, physiotherapist and nurse [14.37% (45/313) of patients].

Passive exercises were more frequently performed [46.3% (145/313) of patients], followed by active [24.6% (77/313) of patients] and active-assisted [19.1% (60/313) of patients]. Passive

**Table 2. Prevalence of mobilization activities and type of exercises performed according to ventilatory support.** Values represent n (%).

| Mobilization activities | All Patients (n = 358) | Mechanically Ventilated (n = 158) | Non-mechanically ventilated (n = 200) |
|---|---|---|---|
| Prevalence of mobilization, n (%) | 313 (87.4) | 135 (85.4) | 178 (89) |
| Type of exercises, n (%) | | | |
| Passive | 145 (46.3) | 112 (82.3) | 34 (19.1) |
| Assisted | 26 (8.3) | 9 (6.6) | 17 (9.5) |
| Active-assisted | 60 (19.1) | 11 (8.1) | 49 (27.5) |
| Active | 77 (24.6) | 2 (1.5) | 74 (41.6) |
| Resisted | 5 (1.6) | 2 (1.5) | 3 (1.7) |
| In-bed exercise, n (%) | | | |
| Passive | 188 (60.1) | 113 (83.7) | 59 (33.1) |
| Out-of-bed exercises, n (%) | 125 (39.9) | 22 (16.3) | 119 (66.8) |
| Passively moved to chair | 8 (6.4) | 2 (9.1) | 6 (5.0) |
| Sitting over the edge of bed | 30 (24.0) | 10 (45.4) | 20 (16.8) |
| Standing | 7 (5.6) | 3 (13.6) | 31 (26.0) |
| Transfering bed to chair | 17 (13.6) | 3 (13.6) | 5 (4.2) |
| Marching on spot | 3 (2.4) | 0 (0) | 11 (9.2) |
| Walking with assistance of 2 or more people | 13 (10.4) | 2 (9.1) | 23 (19.3) |
| Walking with assistance of 1 person | 24 (19.2) | 1 (4.5) | 5 (4.2) |
| Walking independently with a gait aid | 5 (4.0) | 0 (0) | 17 (14.3) |
| Walking independently without a gait aid | 18 (14.4) | 1 (4.5) | 1 (0.8) |

exercises were more common among mechanically ventilated patients than non-mechanically ventilated patients (Table 2). In bed exercises were more frequently performed than out of bed exercises, especially in patients under mechanical ventilation (Table 2). Patients without any ventilatory support were more frequently mobilized out of bed.

In mechanically ventilated patients a total of 202 barriers to achieving a higher activity level were reported. The most frequently reported barrier was due to hemodynamics in patient [34/202; 16.8%], followed by the absence of early mobilization protocol [29/202; 14%] and excessive sedation [25/202; 12.3%]. In those mechanically ventilated patients receiving passive

**Table 3. Multivariable logistic regression model of factors associated with mobilization activities.**

| Variables | OR | 95% CI | p value |
|---|---|---|---|
| SOFA score | 0.95 | 0.84–1.07 | 0.39 |
| Ventilatory support | | | |
| No support | 1.00 | (Reference) | — |
| Invasive mechanical ventilation | 1.46 | 0.45–4.73 | 0.52 |
| Noninvasive ventilation | 1.37 | 0.55–3.42 | 0.49 |
| Use of vasoactive drugs | 0.89 | 0.37–2.14 | 0.80 |
| Type of hospital | | | |
| Public | 1.00 | (Reference) | — |
| Private | 1.04 | 0.27–3.97 | 0.95 |
| University | 0.38 | 0.12–1.16 | 0.09 |
| Number of physiotherapists per 6-hour shift | 0.82 | 0.59–1.12 | 0.21 |
| Number of patients per physiotherapist | 1.09 | 0.93–1.27 | 0.29 |
| Institutional early mobility protocol | 3.19 | 1.23–8.22 | 0.01 |

OR: Odds Ratio, 95% CI: 95% Confidence Interval, SOFA score: sequential organ failure assessment.

**Table 4.  Multivariable logistic regression model of factors associated with out-of-bed exercise.**

| Variables | OR | 95% CI | p value |
|---|---|---|---|
| SOFA score | 0.72 | 0.60–0.86 | < 0.001 |
| Ventilatory support | | | |
| No support | 1.00 | (Reference) | — |
| Noninvasive ventilation | 0.31 | 0.10–0.97 | 0.04 |
| Invasive mechanical ventilation | 0.13 | 0.04–0.41 | < 0.001 |
| Use of vasoactive drugs | 1.18 | 0.34–4.03 | 0.79 |
| Type of hospital | | | |
| Public | 1.00 | (Reference) | — |
| Private | 1.63 | 0.54–4.96 | 0.38 |
| University | 2.81 | 0.79–9.94 | 0.10 |
| Number of physiotherapists per 6-hour shift | 1.17 | 0.90–1.52 | 0.24 |
| Number of patients per physiotherapist | 1.30 | 1.07–1.59 | < 0.001 |
| Institutional early mobility protocol | 5.80 | 1.33–25.30 | 0.02 |

OR: Odds Ratio, 95% CI: 95% Confidence Interval, SOFA score: sequential organ failure assessment.

exercises a total of 190 barriers were reported. The most frequently reported barriers for these patients were the absence of early mobilization protocol [27/190; 14%] and access to specialized equipment [27/190; 14%], followed by hemodynamics in patient [26/190; 13.7%] and excessive sedation [22/190; 11.5%].

Reasons for not performing mobilization were mostly due to contra indications [55.5% (25/45)], followed by barriers related to the absence of an early mobility protocol [26.6% (12/45) patients] and unavailability of physiotherapists [17.7% (8/45) of patients].

## Factors associated with mobilization activities and out-of-bed exercises

Multivariable logistic regression analysis of factors associated with mobilization activity is provided in Table 3. Multi-collinearity was checked for all variables; no collinearity was present (S1 Table in S1 Text). After adjusting for confounders, the only independent predictor of ICU mobilization was the presence of an institutional early mobility protocol (OR, 3.19; 95% CI, 1.23 to 8.22; p = 0.016) (Table 3).

Multivariable logistic regression analysis of factors associated with out-of-bed exercise is provided in Table 4. After adjusting for confounders, independent predictors for out-of-bed exercise were: SOFA score, use of noninvasive ventilation, use of invasive mechanical ventilation, number of patients per physiotherapist, and the presence of an institutional early mobility protocol (Table 4).

Multivariable logistic regression analysis of factors associated with early mobilization and with out-of-bed exercise in patients included within 72 hours of ICU admission is described in S2 Table in S1 Text.

## Safety

Safety events related to mobilization were reported in 8.6% (27/313) of patients, mainly respiratory distress in 59.2% (16/27) of patients and hemodynamic instability in 22.2% (6/27) of patients. Accidental chest tube, central venous catheter, peripheral catheter and chest drain removal were not reported.

## Discussion

The main finding of this 1-day prospective multicenter point prevalence study was that approximately 90% of critically ill patients treated in Brazilian ICUs received mobilization therapy. Moreover, the presence of an institutional early mobility protocol was associated with a threefold higher chance of ICU mobilization during that day.

The vast majority of patients receiving mechanical ventilation included in our study received passive mobilization. Our results, in the mechanically ventilated patients, are in agreement with the results reported in a recent study performed in 11 ICUs located in southern Brazil [18]. Nevertheless, the prevalence of mobilization in mechanically ventilated patients found in our study was higher than the prevalence between 32 to 45% reported by other authors [15–18]. We believe that the high prevalence of mobilization in mechanically ventilated patients found in our study may be explained, at least in part, by the fact that, in Brazil, physiotherapists are part of the multidisciplinary ICU team assisting critically ill patients throughout the ICU stay.

In our study, the average SOFA score was very low. This finding is probably due to the fact that most patients did not require invasive mechanical ventilation at the study day, which influences the SOFA score. The SOFA score was measured for the study day in order to correlate with the mobilization practice. Another aspect to consider is that patients were included on any day during ICU stay, with the majority of patients on the sixth day of ICU admission. They may have been included during an improvement in their clinical setting. This finding may also have an impact on the prevalence of mobilization found in the present study. Similar ICU length of stay, with a median (IQR) of 7 (3–7) days, was also described by Fontela and colleagues [18]. In Brazil, as reported by the Brazilian Intensive Medicine Association, the mean ICU length of stay in 2017 was 16 days. Most Brazilian hospitals do not have step down units. As a result, patients may stay in the ICU longer in order to be clinically stable before receiving ward discharge.

Another important aspect, which differs from ICUs in the USA, but is similar to many ICUs throughout the world, is that Brazilian physiotherapists are part of the ICU team and are responsible for both respiratory and mobilization therapy of critically ill patients. As a result, regarding patients that already have a respiratory therapy prescription, such as mechanically ventilated patients, physiotherapists, since they have an independent practice, can decide when to start mobilization, which may explain the high prevalence of mobilization in these patients. In most ICU patients, the usual decision making to initiate mobilization in most Brazilian ICUs is related to the physician evaluation after ICU admission, in which case the decision to start takes place earlier than the physiotherapy evaluation. All the mobilization events reported in our study were led by physiotherapists; similar results were described by Fontela and colleagues [18]. Quality improvement studies suggest that dedicated ICU therapists enhance access to mobilization [22,23]. Similar results were found in a randomized study with an early involvement of physiotherapists and occupational therapists in mechanically ventilated patients [12]. Another finding in our study similar to the Fontela and colleagues' study [18], is the nurse to patient ratio of 1:6, which is higher than many international ICUs. Although this is in accordance with the Brazilian Federal Nursing Council, nurses assume a more managerial position having a nurse assistant to deliver patient care managed by the nurses. Due to our national nursing practice, physiotherapists are the ones responsible for performing most of the mobilization therapy, while nurses and nurses' assistants may help with patient mobilization.

The presence of an institutional early mobility protocol was reported only in 57.7% of ICUs included in our study. This result was similar to those observed in the United States (53%)

[17], yet lower than what was observed in Germany (71%) [15]. The absence of an early mobility protocol is considered a structural barrier for mobilization [24]. Without a mobility protocol, the ICU team will not be able to identify the safety criteria to start mobilization nor a standardized protocol to be followed by all team members. Previously published studies [11–14] have also shown that the presence of an early mobility protocol, when compared to usual care, improves hospital length of stay, mechanical ventilation duration, and delirium. In addition, patients get out of bed earlier. We found that the presence of an early mobility protocol was positively associated with mobilization and with out-of-bed exercises, in accordance with these previously randomized published studies.

The mobility protocols in the included ICUs usually start with a patient evaluation and clinical criteria to start mobilization, such as hemodynamic and respiratory reserve and without any contraindications. Based on this evaluation patient may be included in one of the 4 phases of mobilization as previously published and recommended [11,12,21].

Passive exercises were the most frequently performed type of exercise in mechanically ventilated patients in our study. The prevalence of passive exercises in mechanically ventilated patients reported in the USA prevalence study was 62%, while our prevalence was 82.3% [17]. This discrepancy may be related to the fact that the involvement of physiotherapist in patient's mobilization included in the USA study was lower than our study, with only 20% involvement of physiotherapist compared to 84.6% respectively. The barriers to achieving a higher activity level in mechanically ventilated patients reported in our study may also be responsible for a higher prevalence of passive exercises reported in this population. In our study we found that the use of invasive and noninvasive mechanical ventilation was inversely associated with out-of-bed exercises, in accordance with the USA study [17], and it was also reported as a barrier in the Fontela and colleagues' study [18].

Safety events related to mobilization activities were reported in 8.62% of patients, which differs from data reported on Germany (21%), Australia and New Zealand (5%), and the United States (0.9%) [15–17]. Most of the safety events were related to respiratory distress. Similar findings were reported by Nydahl and colleagues [10]; however, in their study, safety events were reported in 2.6% of mobilizations. Unfortunately, our study has no data on how long these safety events lasted and the interventions required.

Our study has limitations. First, in order to include more ICUs around Brazil and be able to understand the practice of early mobilization in our country, a total of 100 ICUs were invited from a total of 1291 ICUs in Brazil, of which only 26 ICUs participated in the study, representing at least one ICU from each Brazilian state. The ICUs included represent those invited in terms of type and size. Secondly, this was a one-day observational study, in which a specific date was established for data collection. Nevertheless, in order not to influence the health care team in mobilization activities on the study day, the researchers responsible for each site were previously informed of the study day but oriented to hide this information and keep the health care team blinded to the study's objectives. The researchers were responsible to collect the data. Thirdly, there is a lack of information on the severity and duration of adverse events.

## Conclusion

In this nationwide one-day point prevalence study in 26 Brazilian ICUs we found a high prevalence of mobilization activities in critically ill patients; however, there was almost no active mobilization in the mechanically ventilated patients. Moreover, the presence of an institutional early mobility protocol was associated with a threefold higher chance of ICU mobilization during that day.

The impact on outcomes of early mobilization of critically ill patients admitted to Brazilian ICUs needs to be further addressed.

## List of all authors of e-MOTION investigators

Cintia MC Grion (Hospital Universitário Regional do Norte do Paraná), Edmilson L B de Moura (Hospital Santa Luiza), Clara Gaspari (Instituto Estadual do Cérebro Paulo Niemeyer), Anna Carolina Jaccoud (Instituto Estadual do Cérebro Paulo Niemeyer), Karina T. Time-netsky (Hospital Israelita Albert Einstein), Thiago D. Corrêa (Hospital Israelita Albert Einstein), Ary Serpa Neto (Hospital Israelita Albert Einstein), Raquel AC Eid (Hospital Israelita Albert Einstein), Renato C. de Freitas Chaves (Hospital Israelita Albert Einstein), Denise Car-nieli Cazati (Hospital Israelita Albert Einstein), Wellington P Yamaguti (Hospital Sírio-Liba-nês), Morian Akemi Onoue (Hospital Sírio-Libanês), Ana Lígia Vasnconcellos Maida (Hospital Sírio-Libanês), Carolina M Pellegrino (Hospital Geral do Grajaú), Monique But-tignol (Hospital Municipal Vila Santa Catarina), Renata H Moura (Hospital Municipal Vila Santa Catarina), Eliana B Caser, Betania S Sales (Hospital Unimed Vitória), André Gobatto (Hospital da Cidade), Cristina P Amendola (Hospital de Câncer de Barretos–Fundação Pio XII), Jonathas J da Silva (Hospital de Câncer de Barretos–Fundação Pio XII), Vandack Nobre (Hospital das Clínicas da Universidade Federal de Minas Gerais), Lídia Mourão Barreto (Hospital das Clínicas da Universidade Federal de Minas Gerais), Cintia Mora (Hospital Ministro Costa Cavalcanti), Leandro Taniguchi (Hospital das Clínicas da Faculdade de Medicina da USP),Vivian Sales (UTI do Departamento de Moléstias Infecciosas), Evelin Cechinatti (Hospital Estadual Américo Brasiliense), Cezar Luz, Adriana Toma (Hospital Estadual Américo Brasiliense), Jorge Paranhos (Santa Casa de Misericórida–São João Del Rei), Adilson Carvalho (Santa Casa de Misericórida–São João Del Rei), Louise AR Gondim (UDI Hospital Empreen-dimentos Medico Hospital Do Maranhão), Lanese M de Figueiredo (Hospital Distrital Evandro Ayres de Moura), Márcio Duarte (Hospital Municipal Evandro Freire), Gleiciana Vargas (Hospital Municipal Evandro Freire), Aline Santos (Hospital Municipal Evandro Freire), Michele Godoy (Hospital das Clínicas da Universidade Federal de Pernambuco), Fabianne Dantas (Hospital das Clínicas da Universidade Federal de Pernambuco), Paulo C N Fortes (Hospital Regional do Sudoeste Walter Alberto Pecóits), Raimundo Nonato (Hospital do Cor-ação Anis Rassi), Paula Vassalo (Hospital Universitário Cassiano Antônio Moraes–Universi-dade Federal do Espírito Santo), Márcia M P Dantas (Instituto Doutor José Frota), Lenise Fernandes (Instituto Doutor José Frota), Giovanna Carvalho (Hospital Universitário Pedro Ernesto–Universidade do Estado do Rio de Janeiro), Sergio Cunha, Mônica Cruz (Hospital Universitário Pedro Ernesto–Universidade do Estado do Rio de Janeiro).

## Supporting information

**S1 Text.**
(DOCX)

## Acknowledgments

We want to express our gratitude to all the participating ICUs of this study. The authors thank Helena Spalic for proofreading this manuscript.

## Author Contributions

**Conceptualization:** Karina T. Timenetsky, Ary Serpa Neto, Murillo S. C. Assunção, Raquel A. C. Eid, Thiago D. Corrêa.

**Data curation:** Karina T. Timenetsky, Ary Serpa Neto, Thiago D. Corrêa.

**Formal analysis:** Karina T. Timenetsky, Ary Serpa Neto, Thiago D. Corrêa.

**Investigation:** Karina T. Timenetsky, Leandro Taniguchi, Thiago D. Corrêa.

**Methodology:** Karina T. Timenetsky, Ary Serpa Neto, Murillo S. C. Assunção, Raquel A. C. Eid, Thiago D. Corrêa.

**Project administration:** Karina T. Timenetsky.

**Resources:** Murillo S. C. Assunção, Leandro Taniguchi, Raquel A. C. Eid.

**Supervision:** Karina T. Timenetsky, Thiago D. Corrêa.

**Writing – original draft:** Karina T. Timenetsky, Ary Serpa Neto, Murillo S. C. Assunção, Leandro Taniguchi, Raquel A. C. Eid, Thiago D. Corrêa.

**Writing – review & editing:** Karina T. Timenetsky, Ary Serpa Neto, Murillo S. C. Assunção, Leandro Taniguchi, Raquel A. C. Eid, Thiago D. Corrêa.

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
