## [Decision Letter · Decision Letter 0]

10 Dec 2019

PONE-D-19-29730

EARLY MOBILIZATION IN THE ICU: A 1-DAY POINT- PREVALENCE NATIONWIDE STUDY IN BRAZIL

PLOS ONE

Dear Timenetsky,

Thank you for submitting your manuscript to PLOS ONE. After careful consideration, we feel that it has merit but does not fully meet PLOS ONE’s publication criteria as it currently stands. Therefore, we invite you to submit a revised version of the manuscript that addresses the points raised during the review process.

In my personal opinion this manuscript contains many concerns and the statistical analysis is not well and rigorously performed. I think that with a rigorous revision the manuscript could be improve. 

Reviewers produced very well performed revisions, I would like you to try to answer to all the questions, specifically with a particular attention to the statistical analysis. There are non conflicts between the reviews

We would appreciate receiving your revised manuscript by dec 30th . To enhance the reproducibility of your results, we recommend that if applicable you deposit your laboratory protocols in protocols.io, where a protocol can be assigned its own identifier (DOI) such that it can be cited independently in the future. For instructions see: http://journals.plos.org/plosone/s/submission-guidelines#loc-laboratory-protocols

We look forward to receiving your revised manuscript.

Kind regards,

Martina Crivellari

Academic Editor

PLOS ONE

Journal Requirements:

2. Please include captions for your Supporting Information files at the end of your manuscript, and update any in-text citations to match accordingly. Please see our Supporting Information guidelines for more information: http://journals.plos.org/plosone/s/supporting-information

Reviewers' comments:

Reviewer's Responses to Questions

**Comments to the Author**

1. Is the manuscript technically sound, and do the data support the conclusions?

Reviewer #1: Partly

Reviewer #2: Partly

2. Has the statistical analysis been performed appropriately and rigorously? 

Reviewer #1: No

Reviewer #2: Yes

3. Have the authors made all data underlying the findings in their manuscript fully available?

Reviewer #1: No

Reviewer #2: No

4. Is the manuscript presented in an intelligible fashion and written in standard English?

Reviewer #1: Yes

Reviewer #2: Yes

5. Review Comments to the Author

Reviewer #1: Thank you for the opportunity to review this manuscript that deals with the important issue of mobilisation of ICU patients. I do have some substantial comments that I believe require consideration and potential revision of the manuscript as outlined below:

In the discussion the authors mention that ‘we included patients within the first 24 hours of mechanical ventilation’ however there is nothing in the methods to suggest only patients in the first 24 hours of mechanical ventilation were enrolled. Consequently, I am assuming that patients might have been on any day of their ICU stay when they were included in the study - this should be made clear. Further, it would be beneficial to provide an indication of how many patients were on day 1 day 2, day 3 etc of their ICU stay, otherwise this discussion point (lines 315 – 316) is not valuable as it is not possible to determine if, and how many, day 1 patients were included.

Assuming that my understanding above (that you included patients on any day of their ICU stay) is correct, I do not believe you can use the term ‘early mobilization’ throughout the manuscript as you have not measured ‘early’ mobilization, only ‘any’ mobilization in the ICU.

Lines 164 – 165: why was mobilization performed only by a physiotherapist, rather than by any member of the inter-disciplinary team, recorded as occurring? This is particularly important in the context that you discuss the importance of the inter-disciplinary team in your discussion, and at the very least needs to be included as a study limitation. This also impacts on your discussion (lines 305 – 306)

Your findings indicate that half of the mobilisation, and almost all of the mobilisation in the mechanically ventilated patients, was limited to passive exercises – this is an important finding and needs to be conveyed in the abstract, results and conclusion, as well as discussed more fully. The result is that although there was a high level of mobilisation, there was almost no active mobilisation in the mechanically ventilated patients.

Page 6 – please clarify that individual patients did not need to provide consent

Line 188 – you indicate that 26 of 100 invited ICUs participated, but do you have data on how many ICUs exist in Brazil? If so, this should be included. Also, did the included ICUs represent those invited and the entire Brazilian cohort of ICUs in terms of type and size? This also needs to be incorporated into the relevant section of your discussion (lines 335 – 337).

Page 9 and elsewhere – the patient to nurse ratio is extremely high for many ICUs internationally, and the average SOFA score is very low; linked to this is that only 44% of patients were mechanically ventilated and a further 12% non-invasively ventilated, leaving 44% not requiring ventilation support at all. This combination of data raises questions about whether the included units were all ICUs or whether some were more representative of what might be referred to as a High Dependency Unit and how this context compares internationally; discussion of this should be included.

Line 206 – does 358 represent all patients who were in the participating ICUs on the study day, or were some excluded and if so how many and why? This information needs to be included in figure 1.

Lines 235 – 237: it would be useful to understand the usual decision making in the practice setting, i.e. do physios normally have independent practice or is it a policy requirement that physicians need to make the decision about who and when to mobilise? Linked with this is detail regarding the early mobility protocols that you refer to – what was normally included in these?

Line 251: can you give more information about the contra-indications and who decided that these were contra-indications?

Lines 251 – 252: it’s not clear to me while the absence of a mobility protocol is a barrier – please clarify.

Tables 3 and 4 – analysis:

- I would expect number of patients per physiotherapist and number of physiotherapists per 6-hour shift to potentially be highly correlated; was multi-collinearity checked for? Similarly, can you confirm that multi-collinearity between type of hospital and SOFA score was checked for?

- It is not clear to me why you have entered ‘invasive mechanical ventilation’ and ‘noninvasive ventilation’ as 2 separate factors in the regression, rather than ‘type of ventilation’ being the factor with 3 possible response categories (invasive, non-invasive, no ventilation support – this latter category should be the reference category)

- Type of hospital – given you have OR for 3 different categories it is not clear to me what the reference category was – can you clarify?

Lines 285 – 287: did you look at how long the safety events lasted, and whether any intervention was required to resolve the safety issue?

Lines 324 – 327 – I am a little uncertain how the number of students available might affect mobilisation rates when you only considered mobilisation performed by a physiotherapist – please clarify.

Minor comments:

- Please write ‘colleagues’ out in full rather than ‘cols’

Reviewer #2: This is a very promising manuscript presenting mobilization data of a 1-day prevalence study in Brazil. Before accepting the manuscript, however, several major points should be clarified in my point of view.

Major points:

A. This prevalence study is not investigating early mobility or early mobilization because it investigates all ICU patients regardless of the length of the ICU stay. Therefore, the manuscript should be rephrased accordingly and “early” should be omitted throughout. Furthermore, the term mobilization or mobilization therapy should be used instead of mobility, because the patients are mobilized mainly.

B. That said, it would be interesting to see early mobilization data. Do the authors have the ICU day of the patients so that they can provide data on early mobilization (e.g. within 72 hours of ICU admission)?

C. You define your “mobilization therapy” by the ICU mobility scale. The ICU mobility scale, however, is a mobilization therapy strategy only incorporating active mobilization forms, while you use passive, active-assisted and active forms. Please change your definition accordingly. Since you do not use the IMS in the presentation of the data, you could omit it? Otherwise, describe your changed definition.

D. Furthermore, I have strong objections of your definition of in- and out-of bed exercises. Sitting (passively in a chair) and sitting at the edge (IMS 3) of bed is a typical out of bed exercise while passive mobilization, sitting in the bed and exercises like bedergometry are typical in-bed exercises (ICU Mobility Scale = 1). Please provide references which have used the definitions you used or change your presentation of the data accordingly.

E. How many ICUs are there in Brazil (approximately)?

F. Please provide the study protocol as supplement.

G. Please provide the appropriate checklist from the https://www.equator-network.org/ as supplement.

H. You have not provided your dataset as required by PLOS One.

Because of these major points the result section and discussion section will have to go substantial changes and will have to be assessed then. For the discussion section, I think the comparison with the other 1-day prevalence studies and differences to their results should be discussed in more detail, since the methods were not identical. Consequently, the prevalence numbers cannot be compared without highlighting the differences.

Minor points:

Background:

1. The authors should reconsider references 5-9 for the statement line 109/110. There is a recent metanalysis for example available (https://doi.org/10.1371/journal.pone.0223185) addressing this point

2. The references 10-13 should be reconsidered as well. Please be accurate with mobilization RCTs which improved independent physical status at hospital discharge, which ones improved ventilation days, etc.

3. Please change “and cols.” to “et al.” through the manuscript

Methods

1. Please provide the date of the prevalence study.

2. Please omit reference 18. Redcap does not need 2 references.

3. Please explain what 6-hour shift means. Does that mean that in Brazil you have physical therapy during 24 hours and you have 4 shifts of 6 hours? Is the ratio unvaried over the day?

4. P. 7 line 160/161: “related to the study day”. Please rephrase since ICU admission diagnosis was not.

5. You use SOFA and need of vasopressors as confounders. Since vasopressors are in the SOFA score, please check for collinearity and provide information.

6. Provide a definition of “sedation” (used in table 1) in the methods section

Discussion

1. P. 14 line 295: change “during the ICU stay” to “that day” or “on XX / XX / 20XX)

2. Line 313-314: In that case I believe that your very good physical therapy staffing might have an influence, since passive mobilization is done by them. I would consider bringing up that argument there.

3. Please omit reference 22.

4. Line 324: Please change to “University hospitals in Brazil…” if this statement is really true, that students support mobilization therapy at your ICUs. This is not a general truth.

5. Limitations: The generalizability depends on the total number of ICUs in Brazil. See question above.

6. Please explain the blinding to the team, since the patients had to give their consent (how was that not seen by the team). Or was the consent waived by the IRB (if so, please change the statement in the methods section accordingly).

6. PLOS authors have the option to publish the peer review history of their article (what does this mean?). If published, this will include your full peer review and any attached files.

Reviewer #1: No

Reviewer #2: Yes: Friedrich Kuhn & Prof. Dr. Stefan J Schaller

---

## [Author Response · Author response to Decision Letter 0]

24 Jan 2020

We would like to express our gratitude for giving us the opportunity to revise our manuscript. We believe the reviewers made insightful suggestions that have greatly helped address critical unresolved issues to improve the manuscript. 

 All the changes in the paper are highlighted in yellow in the revised version to indicate the revised portions of the manuscript. We made modifications in the manuscript according to the reviewers’ comments keeping it within the word and referenced limits imposed by the Journal. We have also revised the manuscript taking into consideration the request made by the Editorial Office. We expect our manuscript to be suitable now for publication in PLOS ONE.

We have submitted the Response to Reviewers file addressing all the comments made and suggested.

I remain at your disposal for any clarification you might require.

Yours sincerely,

On behalf of all the authors,

Karina T. Timenetsky

---

## [Decision Letter · Decision Letter 1]

25 Feb 2020

PONE-D-19-29730R1

Mobilization practices in the ICU: A nationwide 1-day point- prevalence study in Brazil

PLOS ONE

Dear Dr Timenetsky,

Thank you for submitting your manuscript to PLOS ONE. After careful consideration, we feel that it has merit but does not fully meet PLOS ONE’s publication criteria as it currently stands. Therefore, we invite you to submit a revised version of the manuscript that addresses the points raised during the review process.

 ACADEMIC EDITOR: There are not conflicts between the reviews. The manuscript has technically improved. We  ask you just few minor revisions to make it suitable for publication. 

We would appreciate receiving your revised manuscript by march 9th. To enhance the reproducibility of your results, we recommend that if applicable you deposit your laboratory protocols in protocols.io, where a protocol can be assigned its own identifier (DOI) such that it can be cited independently in the future. For instructions see: http://journals.plos.org/plosone/s/submission-guidelines#loc-laboratory-protocols

We look forward to receiving your revised manuscript.

Kind regards,

Martina Crivellari

Academic Editor

PLOS ONE

Reviewers' comments:

Reviewer's Responses to Questions

**Comments to the Author**

1. If the authors have adequately addressed your comments raised in a previous round of review and you feel that this manuscript is now acceptable for publication, you may indicate that here to bypass the “Comments to the Author” section, enter your conflict of interest statement in the “Confidential to Editor” section, and submit your "Accept" recommendation.

Reviewer #1: (No Response)

Reviewer #2: (No Response)

2. Is the manuscript technically sound, and do the data support the conclusions?

Reviewer #1: Yes

Reviewer #2: Yes

3. Has the statistical analysis been performed appropriately and rigorously? 

Reviewer #1: Yes

Reviewer #2: Yes

4. Have the authors made all data underlying the findings in their manuscript fully available?

Reviewer #1: (No Response)

Reviewer #2: No

5. Is the manuscript presented in an intelligible fashion and written in standard English?

Reviewer #1: Yes

Reviewer #2: Yes

6. Review Comments to the Author

Reviewer #1: Thank you for submitting a much improved version of this manuscript. Despite the improvements I do have a small number of remaining queries and suggestions as follows:

Line 122: why were pregnant patients excluded?

Lines 149 – 151: please add in notation of who determined that a contra-indication was present

Line 155: this information belongs in the ‘participant information’ section

Line 190: given you have nursing assistants as well in your environment, the ratio for them should also be reported as the patient to nurse ratio is meaningless in isolation

Line 193: thank you for clarifying that individual patient consent was required; it is highly unusual to have 100% of ICU patients consent to a study (I do not believe I have ever seen it before) so it needs to be made clear in figure 1 that 358 was the total number of patients in study ICUs on the relevant day, and that 358 patients were recruited. In the text I suggest you outline the process used to obtain consent.

Line 199: this is an extremely long ICU LOS, particularly in the context of the very low SOFA – there needs to be further description of the context and usual practices to explain why patients that predominantly do not require invasive mechanical ventilation spend about a week in ICU as this is very different to most international practice and possibly accounts for why your mobilisation rates are high.

Lines 292 – 296: although the role of the physiotherapist might contribute to the mobilisation level you have reported, I suggest that the very low severity of illness and the relatively long ICU length of stay reported in your cohort might contribute as much, if not more, to the opportunity for mobilisation. The influence of these characteristics is discussed much later in the discussion (lines 346 – 352) and needs to be moved earlier to integrate with this discussion.

Lines 297 – 299: I actually disagree with this statement – there are many areas of the world (including UK, some parts of Europe, Australia & New Zealand, possibly others) where the physiotherapists are part of the ICU team responsible for both respiratory and mobilisation therapy; I tend to think you are making your comparison with the USA which is notable for different therapists undertaking different elements of care and not always being part of the ICU team but this does not reflect much of international practice.

Lines 340 – 342 – repetitive of previous page – suggest deleting

Lines 368 – 373: this information repeats what has been provided earlier in the

Limitations: the lack of information about the severity and duration of adverse events should be added as a limitation

Typographical comments:

- Lines 186 – 187 and elsewhere – ‘respectively’ is often better located after the relevant data or at the end of a sentence, for example ‘… delirium protocols were reported in 50% (13/26) and 38.5% (10/26), respectively, of the ICUs’

- Line 334 – should be ‘were’ rather than ‘where’

- Line 360 – delete ‘P’ after ‘Nydahl’

Reviewer #2: Thank you for revising your manuscript; the presentation has improved substantially. Here are my comments:

General comments

- The format of the references must be corrected. PLOS One does not use superscript references.

- I would be very interested to see data on early mobilization (i.e. within 72 hours of ICU admission). You state that 32.6% fall into this category. Please provide information in the appendix similar to table 3 and 4 in the appendix in this cohort

- The main finding of the study – having an early mobility protocol – is part of the first paragraph of the discussion, but not adequately discussed in the discussion section and not mentioned as a conclusion.

- One concern still exists and has to be clarified before that manuscript can be accepted. Both reviewers asked for the numbers of total ICUs so that it can be adequately assessed what the external validity of your findings is. Providing the total number of ICU beds is not satisfactory for that! Accounting the median size of ICUs in your study and use that to calculate the ICU numbers based on the presented number of 45.000 beds, resulting in 2500 ICUs. That would mean that you present data of 1% of ICUs in Brazil. Please provide the accurate number or this estimation in the manuscript to provide the reader with the adequate context and limitation.

- Just to clarify, you answered that there was no patient excluded. So you had a 100% success rate in getting your deferred consent. Is that correct?

Abstract

- Cut the listening of the mobilization levels with %

- Present the main finding – besides frequency of mobilization (especially passive) as you did, present the findings what influences mobilization (early mobility protocol). This is your message! Revise Conclusion accordingly

Introduction

- P. 4, line 86: Please add “, if applied early,” in the sentence, so it reads “it improves, if applied early, independent physical status at hospital discharge…”

Results

- p 8 study population: please shorten the description with main points (e.g. sex and age), referring to table 1 without repetition of all numbers in the text.

- p.12 line 248. Please add a sentence that no collinearity was present.

- p.12/13 line 261-266: It is not necessary to repeat all factors if they are presented in the table. Feel free to state the significant one but do not repeat all please.

Discussion

- p. 14, line 286 please change “The vast majority of patients receiving mechanical ventilation included in our study received mobilization, but mostly no active mobilization was performed

288 in these patients.“ To “The vast majority of patients receiving mechanical ventilation included in our study received passive mobilization.”

- p.13, line 299-302: Sentence has to be revised for better understanding and English grammar

- p. 16 line 340-342: Repetition of p.13

Table 2

- Line 236: “According to” – the O is missing.

7. PLOS authors have the option to publish the peer review history of their article (what does this mean?). If published, this will include your full peer review and any attached files.

Reviewer #1: No

Reviewer #2: Yes: K Friedrich Kuhn & Prof. Dr. Stefan J Schaller

---

## [Author Response · Author response to Decision Letter 1]

9 Mar 2020

We have made the necessary changes to the manuscript as suggested by the reviewers and editorial office. 

Academic Editor of PLOS ONE

 We would like to express our gratitude for giving us the opportunity to revise our manuscript. We believe the reviewers made insightful suggestions that have greatly helped address critical unresolved issues to improve the manuscript. 

 All the changes in the paper are highlighted in yellow in the revised version to indicate the revised portions of the manuscript. We made modifications in the manuscript according to the reviewers’ comments keeping it within the word and referenced limits imposed by the Journal. We have also revised the manuscript taking into consideration the request made by the Editorial Office. We expect our manuscript to be now suitable for publication in PLOS ONE.

I remain at your disposal for any clarification you might require.

Yours sincerely,

On behalf of all the authors,

Karina T. Timenetsky

Reviewer #1: 

Thank you very much for your additional comments and suggestions.

1- Thank you for submitting a much improved version of this manuscript. Despite the improvements I do have a small number of remaining queries and suggestions as follows:

Line 122: why were pregnant patients excluded?

 Response: We excluded pregnant patients due to the possible mobilization limitation that these patients may present related to pregnancy itself. 

2- Lines 149 – 151: please add in notation of who determined that a contra-indication was present

 Response: Thank you for your suggestion. We added a notation of who determined that a contra indication was present as follows:

“Contra indications for mobilization were considered as respiratory, cardiovascular, neurological or other considerations as described in the study published by Hodgson and colleagues (21). Contra indication was considered present after the health care and research team reached consensus on this topic. “ (Lines 157 – 158)

3- Line 155: this information belongs in the ‘participant information’ section

Response: Thank you. We have included this information in the “participant information” as the reviewer recommended. 

4- Line 190: given you have nursing assistants as well in your environment, the ratio for them should also be reported as the patient to nurse ratio is meaningless in isolation

Response: Thank the reviewer for the comments. We have added this information in the result section. Lines:_195-196._____

“The median (IQR) patient to nursing assistant ratio was 2 (2.0-3.0).” This ratio is in accordance to the Nursing Federal Council recommendation.

5- Line 193: thank you for clarifying that individual patient consent was required; it is highly unusual to have 100% of ICU patients consent to a study (I do not believe I have ever seen it before) so it needs to be made clear in figure 1 that 358 was the total number of patients in study ICUs on the relevant day, and that 358 patients were recruited. In the text I suggest you outline the process used to obtain consent.

Response: Thank the reviewer for the comments. In fact, 100% for informed consent is very high, but due to the nature of the study design, where there was no intervention made, patients and families usually have no restrictions in signing the informed consent. 

The process used to obtain consent involved approaching the family member or patient when they met the study inclusion criteria, then explain the objective of the study, and the possible uses of the information obtained from it. The person responsible for the study in each center was responsible for obtaining consent. This information was included in lines 112-116. We have also made changes in Figure 1 as suggested.

6- Line 199: this is an extremely long ICU LOS, particularly in the context of the very low SOFA – there needs to be further description of the context and usual practices to explain why patients that predominantly do not require invasive mechanical ventilation spend about a week in ICU as this is very different to most international practice and possibly accounts for why your mobilisation rates are high.

Response: Our SOFA score represents the study day SOFA and not the ICU admission SOFA and as the majority of patients were included after 72 hours of ICU admission, patients may have improved during ICU stay. Another important point is that most hospitals do not have step down units, and patients may stay in the ICU longer in order to be clinically stable before receiving ward discharge. Similar ICU LOS, with a median (IQR) of 7 (3-7) days was also described by Fontela and colleagues. In Brazil, as reported by the Brazilian Intensive Medicine Association, the mean ICU LOS in 2017 was 16 days. 

7- Lines 292 – 296: although the role of the physiotherapist might contribute to the mobilisation level you have reported, I suggest that the very low severity of illness and the relatively long ICU length of stay reported in your cohort might contribute as much, if not more, to the opportunity for mobilisation. The influence of these characteristics is discussed much later in the discussion (lines 346 – 352) and needs to be moved earlier to integrate with this discussion.

 Response: We agree with the reviewer and have made the necessary changes moving the discussion of SOFA and length of stay earlier in the discussion as suggested. Lines: 298-311.

8- Lines 297 – 299: I actually disagree with this statement – there are many areas of the world (including UK, some parts of Europe, Australia & New Zealand, possibly others) where the physiotherapists are part of the ICU team responsible for both respiratory and mobilisation therapy; I tend to think you are making your comparison with the USA which is notable for different therapists undertaking different elements of care and not always being part of the ICU team but this does not reflect much of international practice.

Response: We agree with the reviewer and have made changes to this statement in the discussion. Lines: 312-313.

9- Lines 340 – 342 – repetitive of previous page – suggest deleting

Response: We agree with the reviewer and excluded this information. 

10- Lines 368 – 373: this information repeats what has been provided earlier in the

Limitations: the lack of information about the severity and duration of adverse events should be added as a limitation

Response: We have removed this information and included the lack of information about the severity and duration of adverse events in the limitation section.

11- Typographical comments:

a) Lines 186 – 187 and elsewhere – ‘respectively’ is often better located after the relevant data or at the end of a sentence, for example ‘… delirium protocols were reported in 50% (13/26) and 38.5% (10/26), respectively, of the ICUs’

Response: Thank the reviewer, we have made the proper corrections in the manuscript.

b) Line 334 – should be ‘were’ rather than ‘where’

Response: Thank the reviewer, we have made the proper corrections in the manuscript.

c) Line 360 – delete ‘P’ after ‘Nydahl’

Response: Thank the reviewer, we have made the proper corrections in the manuscript.

Reviewer #2: 

Thank you very much for your additional comments and suggestions.

1- The format of the references must be corrected. PLOS One does not use superscript references.

 Response: We have made the proper correction related to the format of the references in the manuscript. We have changed to square brackets as recommended. 

2- I would be very interested to see data on early mobilization (i.e. within 72 hours of ICU admission). You state that 32.6% fall into this category. Please provide information in the appendix similar to table 3 and 4 in the appendix in this cohort

Response: Thank the reviewer for the suggestion. We have provided information in the appendix similar to table 3 and 4 in patients within 72 hours as suggested by the reviewer. This information is described in S2 Table and S3 Table. 

3- The main finding of the study – having an early mobility protocol – is part of the first paragraph of the discussion, but not adequately discussed in the discussion section and not mentioned as a conclusion.

Response: Thank the reviewer for addressing this important point. We have discussed this point in the discussion section and included in the conclusion as suggested as well. 

4- One concern still exists and has to be clarified before that manuscript can be accepted. Both reviewers asked for the numbers of total ICUs so that it can be adequately assessed what the external validity of your findings is. Providing the total number of ICU beds is not satisfactory for that! Accounting the median size of ICUs in your study and use that to calculate the ICU numbers based on the presented number of 45.000 beds, resulting in 2500 ICUs. That would mean that you present data of 1% of ICUs in Brazil. Please provide the accurate number or this estimation in the manuscript to provide the reader with the adequate context and limitation.

Response: We included the total number of ICUs in the discussion section as suggested by the reviewers (total number of 1291 ICUs). Lines: 375

5- Just to clarify, you answered that there was no patient excluded. So you had a 100% success rate in getting your deferred consent. Is that correct?

Response: In fact, 100% for informed consent is very high, but due to the nature of the study design, where there was no intervention made, patients and families usually have no restrictions in signing the informed consent. 

6- Abstract

- Cut the listening of the mobilization levels with %

Response: We have made the necessary changes as suggested by the reviewers.

7- 

- Present the main finding – besides frequency of mobilization (especially passive) as you did, present the findings what influences mobilization (early mobility protocol). This is your message! Revise Conclusion accordingly

Response: Thank the reviewers for the suggestion. We have made the necessary change as suggested. 

8- Introduction

- P. 4, line 86: Please add “, if applied early,” in the sentence, so it reads “it improves, if applied early, independent physical status at hospital discharge…”

Response: We added “if applied early” in the sentence as suggested by the reviewers. (Line: 85-86)

9- Results

- p 8 study population: please shorten the description with main points (e.g. sex and age), referring to table 1 without repetition of all numbers in the text.

 Response: We have made the necessary changed as suggested by the reviewers. (lines 199-206)

- p.12 line 248. Please add a sentence that no collinearity was present.

 Response: We have added a sentence as suggested by the reviewers.(Line 251)

- p.12/13 line 261-266: It is not necessary to repeat all factors if they are presented in the table. Feel free to state the significant one but do not repeat all please.

Response: We have made the necessary changed as suggested by the reviewers. (261-266)

10- Discussion

- p. 14, line 286 please change “The vast majority of patients receiving mechanical ventilation included in our study received mobilization, but mostly no active mobilization was performed

288 in these patients.“ To “The vast majority of patients receiving mechanical ventilation included in our study received passive mobilization.”

Response: We have made the necessary changed as suggested by the reviewers. (line: 288-289)

- p.13, line 299-302: Sentence has to be revised for better understanding and English grammar

Response: We have revised the sentence as suggested by the reviewers.(Line: 315-319)

- p. 16 line 340-342: Repetition of p.13

Response: We have excluded this sentence in p.16 as suggested by the reviewers.

11- Table 2

- Line 236: “According to” – the O is missing.

Response: Thank the reviewers for addressing this point. We have made the necessary correction as raised by the reviewers.

---

## [Editor Report · Decision Letter 2]

13 Mar 2020

Mobilization practices in the ICU: A nationwide 1-day point- prevalence study in Brazil

PONE-D-19-29730R2

Dear Dr. Timenetsky,

We are pleased to inform you that your manuscript has been judged scientifically suitable for publication and will be formally accepted for publication once it complies with all outstanding technical requirements.

With kind regards,

Martina Crivellari

Academic Editor

PLOS ONE
---

## [Editor Report · Acceptance letter]

20 Mar 2020

PONE-D-19-29730R2 

Mobilization practices in the ICU: A nationwide 1-day point- prevalence study in Brazil 

Dear Dr. Timenetsky:

I am pleased to inform you that your manuscript has been deemed suitable for publication in PLOS ONE. Congratulations! Your manuscript is now with our production department. 

With kind regards,

on behalf of

Dr. Martina Crivellari 

Academic Editor

PLOS ONE